evolution/computational biology/statistical physics

conditional cooperation, prestige bias, public good game, human cooperation

**Authors for correspondence:**
Balaraju Battu
e-mail: bala@cbcs.ac.in
Narayanan Srinivasan
e-mail: nsrini@cbcs.ac.in

# Evolution of conditional cooperation in public good games

## Balaraju Battu and Narayanan Srinivasan

Centre of Behavioural and Cognitive Sciences, University of Allahabad, Allahabad 211002, India

BB, 0000-0002-1455-8924

Cooperation declines in repeated public good games because individuals behave as conditional cooperators. This is because individuals imitate the social behaviour of successful individuals when their payoff information is available. However, in human societies, individuals cooperate in many situations involving social dilemmas. We hypothesize that humans are sensitive to both success (payoffs) and how that success was obtained, by cheating (not socially sanctioned) or good behaviour (socially sanctioned and adds to prestige or reputation), when information is available about payoffs and prestige. We propose and model a repeated public good game with heterogeneous conditional cooperators where an agent's donation in a public goods game depends on comparing the number of donations in the population in the previous round and with the agent's arbitrary chosen conditional cooperative criterion. Such individuals imitate the social behaviour of role models based on their payoffs and prestige. The dependence is modelled by two population-level parameters: *affinity towards payoff* and *affinity towards prestige*. These affinities influence the degree to which agents value the payoff and prestige of role models. Agents update their conditional strategies by considering both parameters. The simulations in this study show that high levels of cooperation are established in a population consisting of heterogeneous conditional cooperators for a certain range of affinity parameters in repeated public good games. The results show that social value (prestige) is important in establishing cooperation.

## 1. Introduction

The prehistory of humans was shaped by achieving public goods through cooperation rather than solely depending on selfish interests [1–3]. Examples of public goods include food acquisition by hunting in groups, security through participation in the military, and public transport. By their very nature

public goods are non-discretionary: that is, each individual in a group can benefit equally without contributing to it. Clearly, a rational and selfish agent is tempted to free ride rather than contribute to a public good because, in relative terms, free riding gives higher payoffs than contributing. Hence, cooperation may not be established if individuals only imitate the social behaviour of rational selfish individuals. However, in day-to-day life, individuals are not only concerned about the payoff scores of potential role models. They are also influenced by whether those payoffs were earned by socially approved means. In this study, we build a model of cooperation when the social learning of the strategy of individuals depends not only on success but also on the prestige of potential role models.

The conditions under which individuals decide to reciprocate or cooperate have been studied extensively [4–8]. In dyadic interactions, cooperation is achieved even if individuals behave selfishly [8]. For instance, reciprocal cooperation or strategies like 'tit for tat' (TFT) establish cooperation in certain restricted conditions [3]. However, the ability to establish cooperation by TFT becomes questionable when the population size increases and individuals interact with many other individuals simultaneously. It has been observed that there is a possibility of graded strategies other than TFT in those situations where individuals interact with many other individuals simultaneously [9]. In such situations, reciprocal cooperation crucially depends on group size [10]. In addition, continuous strategies may perform better when group size increases [11].

Cooperation can also be achieved among strangers through mechanisms like indirect reciprocity and network reciprocity [12,13]. The establishment of cooperation in these models crucially depends on the past behaviour of the individuals and the possibility of obtaining future benefits. In the context of public goods provision, cooperative action is driven by concerns for group benefits and fairness [14]. Cooperation is also achieved by imposing punishments and incentives against free riding [15,16], by opting out of a public good game (PGG) [6] and by linking indirect reciprocity [12,17].

Public good provision has been studied using PGGs. In the standard PGG, each individual is given an equal endowment and a chance to voluntarily contribute to the public good. The collected endowment is enhanced by a factor of more than one and distributed equally among all the individuals, irrespective of their contributions towards the public good [18]. Typically, in the initial rounds of repeated PGGs with the same agents, a majority of the population donates to the public good. It has been hypothesized that, in general, a majority of the population behaves like conditional cooperators and these individuals are heterogeneous [14,19–21]. A conditional cooperator donates to a PGG if many others donate to the game. One possibility in a repeated PGG with a population of heterogeneous conditional cooperators is that free riding by even a few agents in the initial rounds would trigger negative reciprocity or free riding in subsequent rounds. Thus, no cooperation is established.

A social learning interpretation would suggest that, after each round, each individual adapts to the social behaviour of successful individuals, i.e. the social behaviour of free riders; doing so inhibits cooperation [22]. However, field studies show that in many social dilemma situations, cooperation is established among conditional cooperators, suggesting that individuals do not imitate strategies of role models solely based on their payoffs [23,24]. It has been postulated that individuals gain prestige by improving group benefits [25,26]. It has been observed, moreover, that in novel environments individuals are biased to imitate the social behaviour of high prestige individuals [27,28]. When individuals are aware of both the payoff and prestige of role models, then individuals could incorporate both these factors in their social learning strategies.

Let us consider a repeated PGG in which social learning is based on both payoff and prestige. In a PGG, an individual incurs a certain cost by donating or improving the group's benefit and, in turn, as a consequence, gains prestige. On the other hand, by free riding, an individual does not gain prestige but gains a relatively higher payoff than the donor. Clearly, payoff-based social learning leads to further free riding and prestige-based social learning leads to higher levels of cooperation. However, in the real world, individuals operate with noisy social information. Hence, it is plausible that, in imitating the social behaviour of role models, individuals not only consider the role models' payoffs, especially if such payoffs are obtained by socially unacceptable means, but also consider the latter's prestige.

It is evident that, in social interactions, perfect information about individuals' prestige and payoff is not accessible. Hence, an individual strictly imitates neither the social behaviour of free riders (high payoff score) nor that of altruists (high prestige). The reality lies in between. The proposed model uses both the payoff and prestige of role models in the imitation process. The model considers the following features:

(i) the population consists of heterogeneous conditional cooperators, i.e. each individual either donates or free rides based on the number of cooperative actions in the population and their arbitrary threshold value (conditional cooperative criterion)
(ii) each individual imitates the social behaviour of a role model based on their role model's prestige and payoff scores, with a certain affinity or bias.

We propose that factors (i) and (ii) can provide suitable conditions for the establishment of cooperation among heterogeneous conditional cooperators in repeated PGG.

We introduce two population-level parameters, namely, reputation selection intensity, $\eta$, which describes population affinity towards reputation, and payoff selection intensity, $\beta$, which describes the population affinity towards the payoff of agents. When $\beta > \eta$, agents are more sensitive to the relative payoff than the relative reputation in their imitation strategies. With $\beta < 1$ and $\eta < 1$, the imitation process does not solely depend on the role models' payoff or their prestige values. In the proposed model, we examine the social dynamics of cooperation by varying $\beta$ and $\eta$ and measuring donation rates across generations of repeated PGG. We expect that establishment of cooperation crucially depends on these population affinity parameters along with payoff and reputation values of role models.

## 2. Model

We use a standard public good game (PGG) and assume that the population consists of heterogeneous conditional cooperators [19–21,29]. A conditional cooperator is an agent who cooperates based on a decision rule. Each agent is born with an arbitrary conditional cooperative criterion (CCC), which is drawn from a uniform distribution of [1, N], where N (=100) is the population size. In the model, a conditional agent donates to the PGG if and only if the number of donations in the previous round is more than the agent's CCC. A perfect conditional cooperator $i$ with $CCC = CCC_i$ donates to public good if and only if $(n_d - CCC_i) > 0$, where $n_d$ is the number of donations in the previous round. We assume that, in each round, each agent is aware of both the number of donations in the previous round and its own CCC value. In the current round, an agent $i$ (with $CCC = CCCi$ value) donates to the public good with probability, $w_i$,

$$w_i = \frac{1}{1 + \exp(-(n_d - CCC_i) \times \alpha)}. \tag{2.1}$$

The parameter $\alpha$ controls the agents' sensitivity to the conditional rule. With $\alpha \geq 2$, whenever $(n_d - CCC_i) > 0$, the conditional agent's donation probability is high enough that a donation happens most of the time. So we set $\alpha = 2$. For $(n_d - CCC_i) = 0$, $w_i = 0.5$ and the agent donates 50% of the time. By donating to the PGG, an agent incurs a cost of one unit as its payoff, but increases its own prestige by one unit. By contrast, free riding costs nothing but decreases the agent's prestige by one unit.

In the model, conditional cooperators repeatedly play a standard linear PGG. After each round, the collected endowment is multiplied by a factor of two and the resultant endowment is distributed equally irrespective of an agents' donation to PGG. Within a round, agents' CCC values remain the same. In the initial round, each agent's prestige score is zero and each agent possesses 10 units of endowment. In the initial round, we start with few donations (two). The game starts with donations by a few individuals in the initial round; therefore, the overall payoff of free riders is higher than the payoff of donors and the overall prestige score of the donors is higher than the prestige scores of the free riders. After each generation, agents are aware of other agents' CCC values, reputations and payoff scores, and each agent updates its social behaviour or CCC value by imitating the social behaviour of a randomly paired individual with certain mutations. In the model, a generation consists of 10 rounds of a repeated PGG.

We update population using a pair-wise comparison process [30]. In the proposed model, each agent $i$ matched with a randomly chosen other agent $j$ decides whether to imitate the social behaviour of agent $j$ with a probability $= q_{ij} \times p_{ij}$.

$$q_{ij} = \frac{1}{1 + \exp(-\triangle r_{ji} \times \eta)} \tag{2.2}$$

and

$$p_{ij} = \frac{1}{1 + \exp(-\triangle \pi_{ji} \times \beta)}. \tag{2.3}$$

In equation (2.2), $\triangle r_{ji} = (r_j - r_i)$ is the relative prestige and in equation (2.3), $\triangle \pi_{ji} = (\pi_j - \pi_i)$ is the relative fitness of agents $i$ and $j$. The reputation and payoffs are accumulated in repeated PGGs. The parameters $\beta$ and $\eta$ control the focal agent's sensitivity to the payoff difference and reputation difference of the randomly picked pair. If $q_{ij} \times p_{ij} = 1$, the agent imitates the valuable member's social behaviour. If $q_{ij} > p_{ij}$, the agent is biased towards the prestige of the paired agent, and if $q_{ij} < p_{ij}$, the agent is biased towards the payoff of the paired agent in imitating the social behaviour of that paired agent. We allow each agent to miscopy the successful agent's CCC value with probability 0.1. These mutations are created by adding a random value, which is drawn from a Gaussian distribution with zero mean and s.d. = 5 (max = 50 and min = −50) to the original CCC values, to the updated value of equation (2.2). If the updated CCC value is greater than $N$, it is rounded off to $N$; if the resultant value is negative, it is rounded off to zero. This allows the population to have unconditional free riders (CCC = $N$) and unconditional cooperators (CCC = 0). All the agents are updated simultaneously.

Simulations were performed for each experimental condition (for each set of fixed parameters $\alpha = 2$, $\beta$, $\eta$). For instance, when $\eta(=2) \gg \beta$, agents are more sensitive to relative reputation and when $\beta(=2) \gg \eta$, agents are more sensitive to relative payoffs. With $0.01 < \eta < 1$ and $0.01 < \beta < 1$, agents occasionally make mistakes in imitating social behaviour based on relative payoffs and reputation scores. We measured donation rates, i.e. the percentage of donations in a generation. The donation rate constitutes a proxy measure of cooperation level in the population. We computed the distribution of CCC values, which indicates the composition of population in a given generation. To reduce individual trial variations, the results were averaged over 20 iterations. We also computed asymptotic donation rates by taking the mean of the last 1000 generations out of 5000 generations. There is no difference in results after the first few thousand generations; therefore, we have plotted the results for the first 5000 generations. We also plotted the distribution of CCC values of the population for different experimental conditions in the 5000th generation.

## 3. Results

Simulations were performed using Matlab. The donation rates were measured by counting the number of donations divided by the number of actions in a generation and converting the fraction into a percentage. Figure 1 shows donation rates over 5000 generations with a population size of 100. Each colour-coded trajectory represents the evolution of donation rates for a particular $\beta$ and $\eta$ value. Figure 1a shows donation rates for $\beta = 2$ and for various $\eta$ values. Figure 1b shows donation rates for $\beta = 0.5$ for various $\eta$ values. In general, with $\eta = 2$ and whenever $\eta \geq \beta$ with $\beta < 0.6$, i.e. when the population shows relatively more affinity towards the prestige of role models than the payoffs of role models, the donation rates are high. Cooperation is also observed for $\eta < 1$ and $\beta < 1$ with $\eta \geq \beta$ and $\eta < 0.6$ (figure 1b and figure 2b), i.e. when the imitation process does not entirely depend on the role models' payoffs and their prestige. There is no cooperation observed for $\beta = 2$ and $\eta \leq 2$, i.e. when the population is more strongly biased towards the payoff than towards the prestige of role models (figures 1a and 2a).

Figure 2 also shows donation rates over 5000 generations with a population size of 100. Figure 2a shows donation rates for $\eta = 2$ and figure 2b shows donation rates for $\eta = 0.5$ for various $\beta$ values. Cooperation is high whenever $\eta \geq \beta$ with $\beta < 0.6$. The results in figure 2b show that when $\eta = \beta = 0.5$ and $\beta < \eta \leq 0.6$, the cooperation levels are high.

Figures 3 and 4 show the distribution of CCC values in the 5000th generation with a bin size of 10. Figure 3a shows the distribution of CCC values $\eta = 2$ and $\beta = 2$. The population is neither dominated by lower nor by higher CCC value agents. Figure 3b shows CCC values of the population with $\eta = 2$ and $\beta = 0.1$. In this case, the population is dominated by agents with very low CCC values. Figure 4a shows the distribution of CCC values with $\eta = 0.5$ and $\beta = 0.6$. The distribution is relatively uniform with the population slightly dominated by agents with low and medium CCC value agents. Figure 4b shows the distribution of CCC values with $\eta = 0.5$ and $\beta = 0.4$. The majority of the population is dominated by agents with lower CCC values.

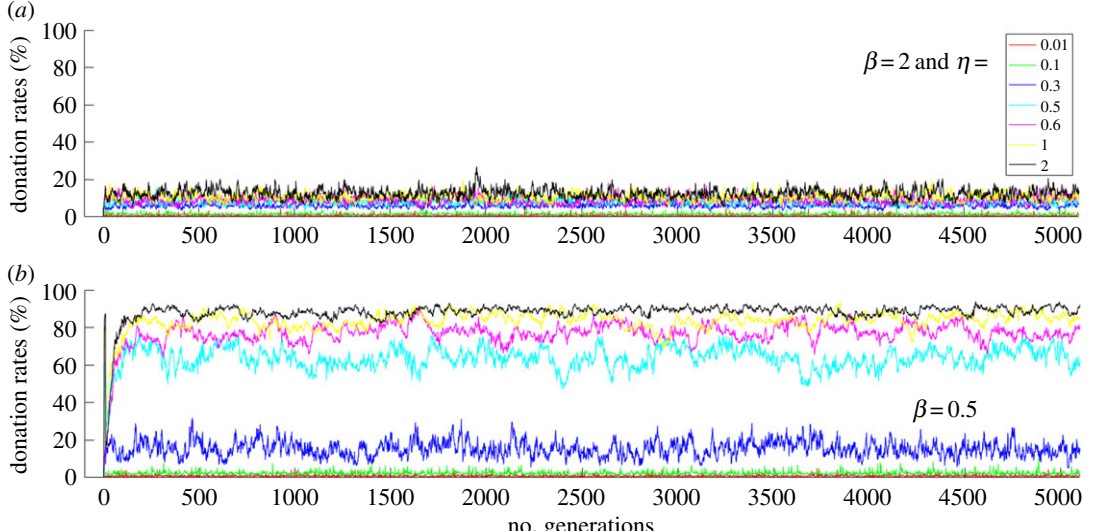

**Figure 1.** Donation rates for (*a*) $\beta = 2$ and for (*b*) $\beta = 0.5$ for various $\eta$ values for 5000 generations.

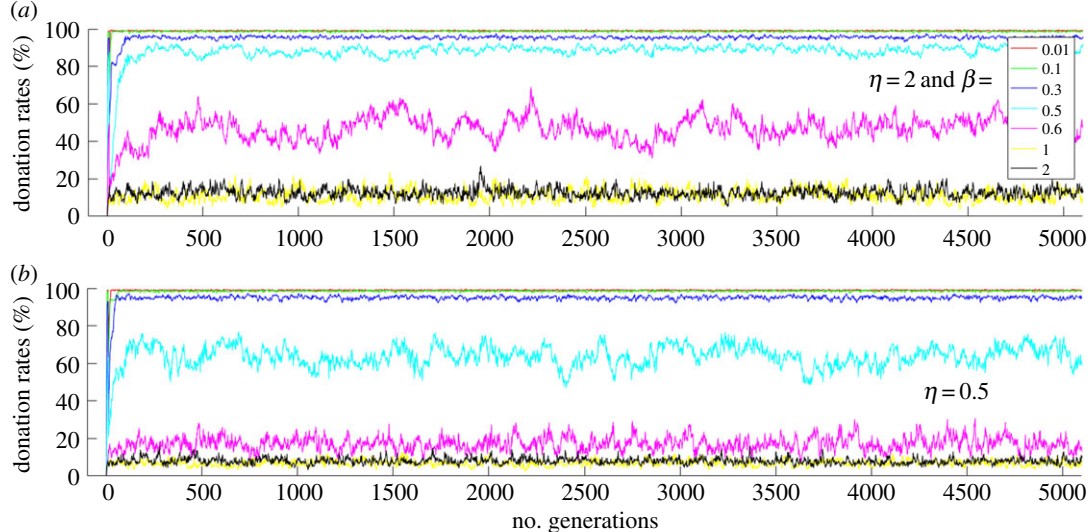

**Figure 2.** Donation rates for (*a*) $\eta = 2$ and for (*b*) $\eta = 0.5$ for various $\beta$ values for 5000 generations.

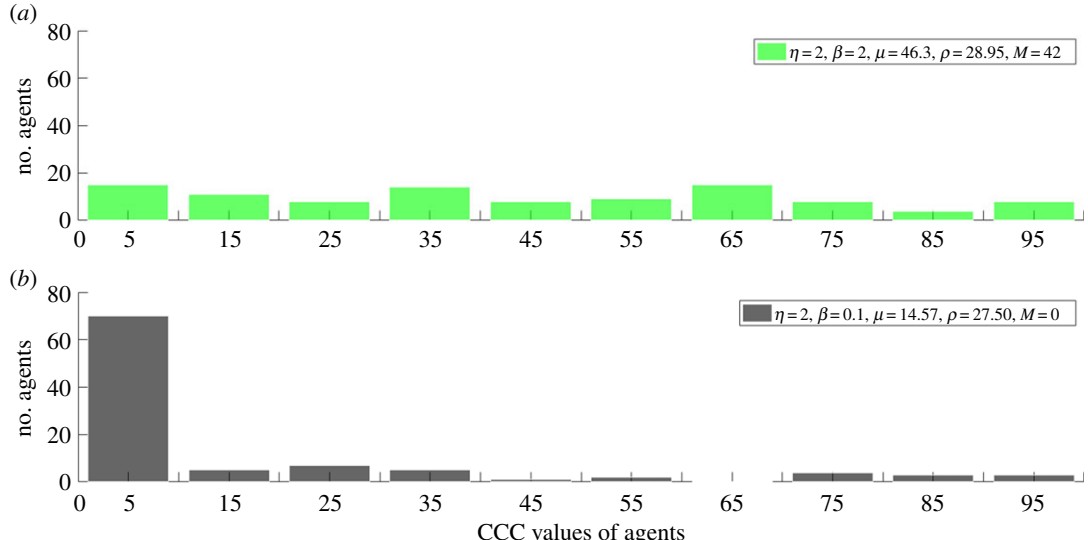

**Figure 3.** The distribution of CCC values in 5000th generation with a bin size of 10 for (*a*) with $\eta = 2$ and $\beta = 2$ and for (*b*) with $\eta = 2$ and $\beta = 0.1$.

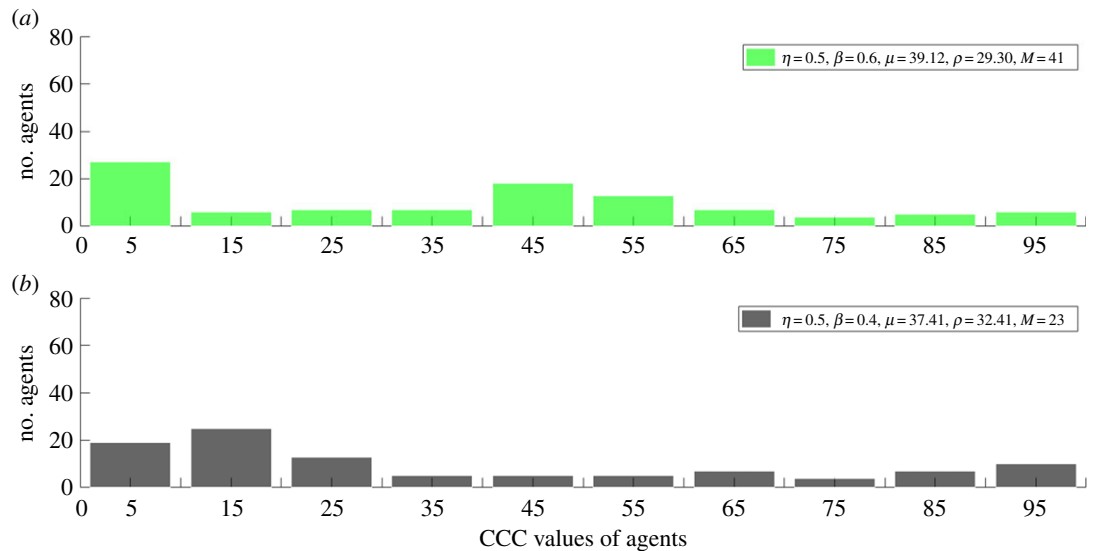

**Figure 4.** The distribution of CCC values in 5000th generation with a bin size of 10 for (*a*) with $\eta = 0.5$ and $\beta = 0.6$ and for (*b*) with $\eta = 0.5$ and $\beta = 0.4$.

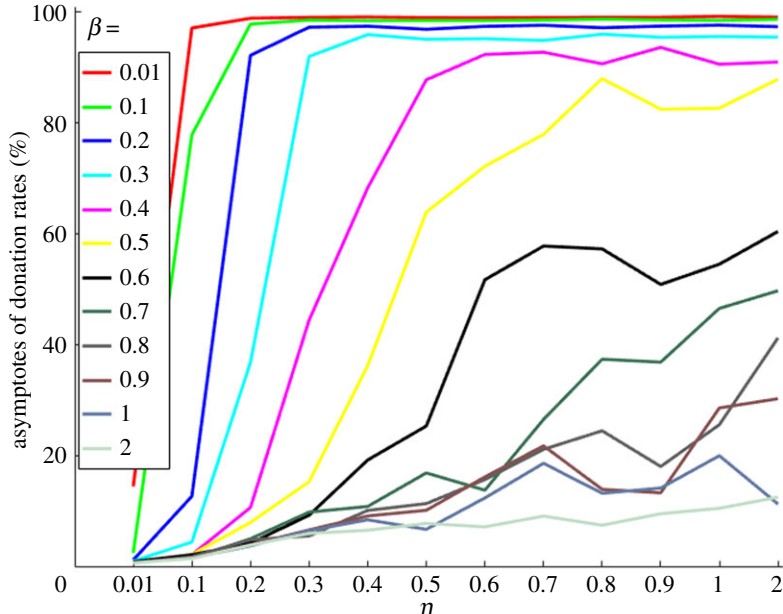

**Figure 5.** Asymptotic averaged donation rates as a function of $\eta$ (x-axis) and $\beta$ (y-axis) values.

Figure 5 shows asymptotic averaged donation rates as a function of $\eta$ and $\beta$ values with each colour-coded trajectory representing asymptotes of donation rates for a particular $\eta$ and $\beta$ value. The donation rates are high both with $\eta \gg \beta$ and also with $\eta = \beta < 1$, indicating that cooperation is achieved not only for $\eta \gg \beta$ but also for lower $\eta$ and $\beta$ ($< 1$).

## 4. Discussion

Stable cooperation is established when heterogeneous conditional cooperators donate to the public good game with their conditional strategies and update their strategies by comparing themselves with role models based on payoff and prestige values. High levels of cooperation are achieved for certain affinity values or parameter settings (figure 5): (i) $\eta \gg \beta$ (see figures 1*b* and 2*a*) and (ii) $\eta \geq \beta$ and $\eta <$ 0.6 (see figures 1*b*, 2*b* and 5). In condition (i), the imitation process is biased towards the prestige of role models compared to the payoff score of role models. In condition (ii), the imitation process depends on both the payoffs and the prestige of the role models. The importance of these results is

that establishing conditional cooperation does not entirely depend on the imitation of role model's prestige. Conditional cooperation crucially depends on reaching a critical amount of cooperation in the population, not just on the conditional strategies of individuals. Both conditions (i) and (ii) provide suitable conditions to establish conditional cooperation when the population is heterogeneous.

The cooperative mechanism proposed in the model is different from the cooperation mechanisms observed in dyadic interactions [31,32] and public good games [17,33–35]. In these studies, a donor's cooperation action depends on the potential receiver's reputation score and individuals reproduce based on relative success. However, in our model, agents cooperate based on conditional strategies and imitate role models' social behaviour not only based on relative payoffs but also based on relative prestige scores. In a population of heterogeneous conditional cooperators, the cooperation observed crucially depends on the conditions required to create a critical amount of cooperation level.

In condition (i), agents are highly biased towards imitating the social behaviour of the high prestige role models. In these conditions, lower CCC agents thrive and higher CCC agents become extinct in the population, and this facilitates the conditions needed to establish high levels of conditional cooperation (figure 2a). In this condition, although the population is highly biased to imitate the social behaviour of high prestige role models, the imitation process of the agent is still influenced by the role model's payoff. As revealed by the histogram of CCC values of population (figure 3b), when individuals are highly biased towards role models' prestige, the population is dominated by agents with lower CCC values. For instance, with $\eta = 2$ and $\beta = 0.1$, median ($M$) of CCC values of population = 0 (figure 3b). In these conditions, cooperation remains high and stable (figure 1b), even if an occasional mutant with CCC = $N$ gets selected into the population. The rest of the agents do not imitate the social behaviour of the mutant because the mutant has a lower prestige score; therefore, cooperation is stable against occasional mutations. In these conditions, since the population is gradually dominated by lower CCC individuals, individuals' payoffs and their prestige scores are similar.

Interestingly, in condition (ii) as well, high levels of stable cooperation are established. When $\eta < 1$ and $\beta < 1$, the imitation process does not entirely depend on the role models' payoff and prestige scores; more often the agents commit errors in their imitation process. For instance, when $\eta \geq \beta$ and $\eta < 0.6$, high levels of cooperation are established in which, in some conditions, the population is slightly more biased towards the prestige of role models. In these conditions, the population remains heterogeneous because neither the role models' prestige nor their payoffs entirely drive the imitation process. In these conditions, the selection process is able to create a critical amount of cooperation levels such that conditional cooperation can be established. In this condition, the population is neither dominated by agents with very high CCC values nor by agents with very low CCC values, but consists of agents with moderate CCC values. This can be seen in figure 4a ($\eta = 0.5$, $\beta = 0.6$, $M = 41$) and figure 4b ($\eta = 0.5$, $\beta = 0.4$, $M = 23$). The manner in which cooperation is established in condition (ii) is different from condition (i). One can interpret condition (ii) as imitation when social information is noisy, that is situations in which agents do not have access to precise prestige or payoff information of role models. The condition is similar to social learning in noisy environments.

In general, the establishment of cooperation by conditional cooperators in a heterogeneous population depends on creating a critical number of cooperative actions in each round. These conditions drive conditional cooperation using nested feedback loops within the generations. It is important to note that certain parameter values of $\beta$ and $\eta$ allow creation of a critical amount of cooperation. Clearly, cooperation is not just established by improving prestige by donations and imitating the most prestigious individual's social behaviour. In both the conditions, cooperation levels are stable even if an occasional mutant with CCC = $N$ who has a high payoff score and a poor prestige score gets selected into the population. In condition (i) these individuals become quickly extinct in the population, because in the population only lower CCC individuals thrive. In condition (ii), individuals with neither higher CCC values nor lower CCC values thrive; only individuals with middle range CCC values thrive.

One can interpret the population parameters $\eta$ and $\beta$ as social values of society. For instance, $\eta$ is much greater than $\beta$ in a value-based society (imitation strongly driven by prestige of successful agents) in which high prestige or reputation agents are treated as role models, and these agents are allowed to participate in future public good games, but agents with poor prestige are not. Therefore, an agent is more likely to get attached to agents with high prestige and imitate their social behaviour. In a competition-based society, imitation is strongly driven by the relative payoff of role models. When $\beta$ is much greater than $\eta$, agents tend to imitate the social behaviour of ruthless competitors who are solely concerned with higher payoff scores and not their prestige. In such a social system, public good provision is not possible. The model shows that heterogeneous conditional cooperators

can achieve high levels of cooperation even if agents do not prefer to imitate only high prestige role models but prefer those with high prestige and high payoff. Perhaps in human societies, individuals who are wealthy and have high prestige are more valuable than those who are only wealthy or those who only have high prestige.

The proposed model does not use existing standard models of cooperation [17,33] and theoretical models of PGGs [36,37]. Instead it crucially depends on the conditional nature of cooperation and imitation based on role models' prestige and payoff scores. In the model, concerns about prestige and reputation are introduced at the imitation stage but not at the donation stage. The role of prestige in our model differs from the role of reputation in models based on indirect reciprocity [15,17] or prestige-based social learning [28].

Further, the model does not use costly punishments against free riders and avoids detrimental effects of punishments [38–40]. The model is limited and does not capture other myriad possibilities of social interactions [31,41]. Despite these constraints, the current model throws light on generic explanations about how conditional cooperation is achieved via prestige-biased social learning and feedback loops.

The model shows how the presence of conditional cooperators in a heterogeneous population in noisy social environments can result in cooperation. The model also points towards social learning mechanisms that are not only driven by payoffs but the mode of obtaining payoffs (when such payoffs are obtained by cheating or socially undesirable means).

Data accessibility. We confirm that all data required are available from the Dryad Digital Repository: https://datadryad. org/stash/dataset/doi:10.5061/dryad.f7m0cfxrc [42].

Authors' contributions. B.B. conceived and designed the study. B.B. programmed the simulation and analysed data. N.S. provided research supervision and guidance to B.B. N.S. helped in data analysis and interpretation. B.B. and N.S. wrote paper. All authors approved the manuscript.

Competing interests. We declare we have no competing financial interest.

Funding. The authors have not received any funding from any agency for this work.

Acknowledgements. We thank Harish Karnick and Laurier Jane Anderson for helping with the manuscript. We also acknowledge the help of the editors and reviewers.

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
