## [Reviewer comments · Royal Society Open Science]

Review History

RSOS-191567.R0 (Original submission)

Review form: Reviewer 1

Is the manuscript scientifically sound in its present form?

Yes

Are the interpretations and conclusions justified by the results?

Yes

Is the language acceptable?

Yes

Do you have any ethical concerns with this paper?

No

Have you any concerns about statistical analyses in this paper?

No

Recommendation?

Accept as is

Comments to the Author(s)

The authors adequately responded to my comments and took them fully into account in this version of the paper.

Review form: Reviewer 2 (Shun Kurokawa)**Is the manuscript scientifically sound in its present form?**

Yes

Are the interpretations and conclusions justified by the results?

Yes

Is the language acceptable?

No

Do you have any ethical concerns with this paper?

No

Have you any concerns about statistical analyses in this paper?

No

Recommendation?

Major revision is needed (please make suggestions in comments)

Comments to the Author(s)

Please see the attached file (Appendix A).

Decision letter (RSOS-191567.R0)

03-Dec-2019

Dear Dr Battu,

The editors assigned to your paper ("Evolution of conditional cooperation in public good games") have now received comments from reviewers. We would like you to revise your paper in accordance with the referee and Associate Editor suggestions which can be found below (not including confidential reports to the Editor). Please note this decision does not guarantee eventual acceptance.

Please submit a copy of your revised paper before 26-Dec-2019. Please note that the revision deadline will expire at 00.00am on this date. If we do not hear from you within this time then it will be assumed that the paper has been withdrawn. In exceptional circumstances, extensions may be possible if agreed with the Editorial Office in advance. We do not allow multiple rounds of revision so we urge you to make every effort to fully address all of the comments at this stage. If deemed necessary by the Editors, your manuscript will be sent back to one or more of the original reviewers for assessment. If the original reviewers are not available, we may invite new reviewers.

To revise your manuscript, log into <http://mc.manuscriptcentral.com/rsos> and enter your Author Centre, where you will find your manuscript title listed under "Manuscripts with

Decisions." Under "Actions," click on "Create a Revision." Your manuscript number has been appended to denote a revision. Revise your manuscript and upload a new version through your Author Centre.

- Data accessibility

If you wish to submit your supporting data or code to Dryad (<http://datadryad.org/>), or modify your current submission to dryad, please use the following link:
<http://datadryad.org/submit?journalID=RSOS&manu=RSOS-191567>

- Competing interests

- Authors' contributions

- Acknowledgements

- Funding statement

Kind regards,

Lianne Parkhouse

Editorial Coordinator

on behalf of Professor Wen-Xu Wang (Associate Editor) and Miles Padgett (Subject Editor)
openscience@royalsociety.org

Reviewers' Comments to Author:

Reviewer: 1

Comments to the Author(s)

The authors adequately responded to my comments and took them fully into account in this version of the paper.

Reviewer: 2

Comments to the Author(s)

Please see the attached file.

Author's Response to Decision Letter for (RSOS-191567.R0)

See Appendix B.

RSOS-191567.R1 (Revision)

Review form: Reviewer 2

Is the manuscript scientifically sound in its present form?

Yes

Are the interpretations and conclusions justified by the results?

Yes

Is the language acceptable?

No

Do you have any ethical concerns with this paper?

No

Have you any concerns about statistical analyses in this paper?

No

Recommendation?

Major revision is needed (please make suggestions in comments)

Comments to the Author(s)

Please see the attached file (Appendix C).

Decision letter (RSOS-191567.R1)

26-Mar-2020

Dear Dr Battu:

Manuscript ID RSOS-191567.R1 entitled "Evolution of conditional cooperation in public good games" which you submitted to Royal Society Open Science, has been reviewed. The comments from reviewer(s) are included at the bottom of this letter.

In view of the criticisms of the reviewer(s), I must decline the manuscript for publication in Royal Society Open Science at this time. However, a new manuscript may be submitted which takes into consideration these comments.

Please note that resubmitting your manuscript does not guarantee eventual acceptance, and that your resubmission will be subject to re-review by the reviewer(s) before a decision is rendered.

You will be unable to make your revisions on the originally submitted version of your manuscript. Instead, revise your manuscript using a word processing program and save it on your computer.

You may also click the below link to start the resubmission process (or continue the process if you have already started your resubmission) for your manuscript. If you use the below link you will not be required to login to ScholarOne Manuscripts.

*** PLEASE NOTE: This is a two-step process. After clicking on the link, you will be directed to a webpage to confirm. ***

https://mc.manuscriptcentral.com/rsos?URL_MASK=7b1ee61c797f4c5296b59f3a94951dd8

Because we are trying to facilitate timely publication of manuscripts submitted to Royal Society Open Science, your resubmitted manuscript should be submitted by 24-Aug-2020. If you are unable to submit by this date please contact the Editorial Office for options.

I look forward to a resubmission.

on behalf of Professor Wen-Xu Wang (Associate Editor) and Miles Padgett (Subject Editor)
openscience@royalsociety.org

Reviewer comments to Author:
Reviewer: 2

Comments to the Author(s)
Please see the attached file.

Author's Response to Decision Letter for (RSOS-191567.R1)

See Appendix D.

Decision letter (RSOS-191567.R2)

22-Apr-2020

Dear Dr Battu,

It is a pleasure to accept your manuscript entitled "Evolution of conditional cooperation in public good games" in its current form for publication in Royal Society Open Science.

The Editors had one or two concerns regarding the degree of advance represented by the paper, but, as Royal Society Open Science does not generally make assessments of the 'impact' of the manuscript, they consider the science to be sound and have recommended publication.

Due to rapid publication and an extremely tight schedule, if comments are not received, your paper may experience a delay in publication. Royal Society Open Science operates under a continuous publication model. Your article will be published straight into the next open issue and this will be the final version of the paper. As such, it can be cited immediately by other researchers. As the issue version of your paper will be the only version to be published I would

advise you to check your proofs thoroughly as changes cannot be made once the paper is published.

on behalf of Professor Wen-Xu Wang (Associate Editor) and Miles Padgett (Subject Editor)
openscience@royalsociety.org

Appendix A

Comment on “Evolution of conditional cooperation in public good games”

Summary:

The authors made a model for public good games, and in the model setting, the evolution of cooperation is possible.

General evaluation

Unfortunately, in the reply, the authors failed in convincing the reviewer 1 that this paper is worthwhile publication. Major revision is required. More concrete comments are below.

Comments

(1) Content

The reviewer 1 points out that the evolution of cooperation could have been predicted in advance from the following two assumptions. (i) Individuals tend to imitate the social behaviour of their most prestigious peers (ii) The prestige of individuals increases when they cooperate. In the reply, the authors wrote "one of the important results of the model is that whenever agents are slightly biased towards high prestige agents, the population is able to create conditions, i.e., critical number of cooperative actions, to establish cooperation." The authors are trying to insist that even when the effect of (i) is small, the evolution of cooperation, so the result remains interesting. Is this correct? A good way to convince the reviewer that the paper is interesting is to remove the effect of (i) "completely". But, the authors say "slightly". It means that the effect of (i) is still required. Do the authors say so? If so, then as the reviewer 1 mentioned, from the assumptions (i) and (ii), the result (i.e., the evolution of cooperation) came easily, and this model is not interesting. On the other hand, if the evolution of cooperation is possible even if the authors completely remove the effect of (i), then the criticism by the review 1 does not hold any more. So, please let me know what happens in the absence of the effect of (i).

The authors wrote, "In our view, the proposed mechanism is novel in comparison to the existing models of cooperation." as a reply to the review 1's comment. I guess that the authors do not understand the review 1's intension. The reviewer 1 is not saying that this submitted model in this paper is similar with a previous work. The reviewer 1 is saying that this submitted model is useless, because the result is easily expected from the two assumptions the authors put. So, even if the authors say that this model is novel, it does not satisfy the reviewer 1 because this is not reviewer 1's point. I

recommend that the authors should read the reviewer 1's comment carefully.

(2) Reference

In the introduction, the authors wrote about repeated PGG. Nonetheless, the authors did not cite Joshi (1987) or Boyd & Richerson (1988). If you know these papers and you chose not to cite them, then I would like to know the reason why you did not cite them. If you did not know them simply, then please read them and think about whether you cite them or not. In addition, Takezawa & Price (2010) is also relevant.

(3) English

The language needs improving. I take one example, but this is just an example, and the authors have to pay more attention for the usage of English. In Introduction section, the authors wrote " In other words, after each round if each individual aware of each other' payoff, the individual adapt ~". Here, how to use "aware" is wrong. On the other hand, immediately after that, the authors wrote "However, if individuals **are** aware of each individual's payoff and their prestige, perhaps individuals more likely to imitate ~". Here, how to use "aware" is correct. So, it seems that the authors know how to use "aware", but the authors are just careless. I recommend that the authors should read this manuscript carefully again. If the authors make this effort, then this paper will become better.

References

- Boyd, R., & Richerson, P. J. (1988). The evolution of reciprocity in sizable groups. *Journal of Theoretical Biology*, 132, 337-356. [http://dx.doi.org/0.1016/S0022-5193\(88\)80219-4](http://dx.doi.org/0.1016/S0022-5193(88)80219-4)
- Joshi, N. V. (1987). Evolution of cooperation by reciprocation within structured demes. *Journal of Genetics*, 6, 69-84. <http://dx.doi.org/10.1007/BF02934456>
- Takezawa, M., & Price, M. E. (2010). Revisiting "The Evolution of Reciprocity in Sizable Groups": Continuous reciprocity in the repeated n -person prisoner's dilemma. *Journal of Theoretical Biology*, 264, 188-196. <http://dx.doi.org/10.1016/j.jtbi.2010.01.028>

Appendix B

Response to Reviewers

Reviewer 1

We thank the Reviewer for accepting the paper.

Reviewer 2

We thank Reviewer 2 for the three major comments. We respond to each of these comments below.

Comment

(1) Content

The reviewer 1 points out that the evolution of cooperation could have been predicted in advance from the following two assumptions. (i) Individuals tend to imitate the social behaviour of their most prestigious peers (ii) The prestige of individuals increases when they cooperate. In the reply, the authors wrote "one of the important results of the model is that whenever agents are slightly biased towards high prestige agents, the population is able to create conditions, i.e., critical number of cooperative actions, to establish cooperation." The authors are trying to insist that even when the effect of (i) is small, the evolution of cooperation, so the result remains interesting. Is this correct? A good way to convince the reviewer that the paper is interesting is to remove the effect of (i) "completely". But, the authors say "slightly". It means that the effect of (i) is still required. Do the authors say so? If so, then as the reviewer 1 mentioned, from the assumptions (i) and (ii), the result (i.e., the evolution of cooperation) came easily, and this model is not interesting. On the other hand, if the evolution of cooperation is possible even if the authors completely remove the effect of (i), then the criticism by the review 1 does not hold any more. So, please let me know what happens in the absence of the effect of (i).

The authors wrote, "In our view, the proposed mechanism is novel in comparison to the existing models of cooperation." as a reply to the review 1's comment. I guess that the authors do not understand the review 1's intension. The reviewer 1 is not saying that this submitted model in this paper is similar with a previous work. The reviewer 1 is saying that this submitted model is useless, because the result is easily expected from the two assumptions the authors put. So, even if the authors say that this model is novel, it does not satisfy the reviewer 1 because this is not reviewer 1's point. I recommend that the authors should read the reviewer 1's comment carefully.

Response:

The basic argument made by Previous Reviewer 1 and current Reviewer 2 is that if statements (1) and (2) are true, then cooperation is obvious. We politely disagree. Perhaps we were not able to communicate our method, results and their implications and we will attempt to do our best here. Our results itself are indicative that this is not necessarily true.

- (a) First, we would like to point out that we are not studying the effect of prestige separately. We are studying the evolution of cooperation when both prestige and payoffs are taken into consideration. Payoff for a particular individual is reduced if that individual cooperates and

prestige increases. In terms of population updating, two parameters influence imitation (product of p_{ij} and q_{ij}) with q_{ij} dependent on prestige and p_{ij} dependent on payoffs. The parameters η and β indicate sensitivity to prestige and payoffs respectively.

- (b) Second, the decision to cooperate is NOT based on the properties of the individual with whom an agent is interacting but based on comparing their own CCC value with a population parameter (number of donations in the previous round in the population). Essentially they are comparing themselves with what is generally happening in the population. If the interactions are dyadic and decision is dependent on the prestige of the individual with whom one interacts, then the reviewer would be correct. However, this is not the case in our model.
- (c) If the reviewer is right and everybody imitates only those with high prestige, we should end up with a population of agents with high prestige. From the perspective of CCC values, we should end with those with low CCC values since they would be more prone to cooperate all the time and in the process will end up with high prestige. If we look at Figure 4, we can see that the distribution of CCC values is spread out. The population consists of agents with low CCC values but also those with high CCC values and medium CCC values (see especially figure 4A). Those with very high CCC values would rarely cooperate (hence would have less prestige) but still there is evolution of cooperation. We do not think it obviously follows from the two hypotheses mentioned by the reviewer. The reviewer would be right if payoffs are not considered at all or those with prestige are rewarded explicitly with payoffs. All is needed are enough agents with low CCC values to drive the population towards significant cooperation under certain conditions (sensitivity to η and β). As the individuals are conditional co-operators and the population is heterogeneous, the amount of cooperation crucially depends on reaching a critical amount of cooperation, not just simply on the strategies of individuals.
- (d) Removing (1) will make people not sensitive to prestige at all and in that case the evolution of cooperation would simply depend on payoffs (which of course has been studied earlier and does not add anything of value). Even with low values of η , there is still a possibility of cooperation provided that the agent is not highly sensitive to payoffs.
- (e) More importantly, cooperation is not always established when individuals imitate most prestigious role models. Cooperation is established for a certain range of η and β values. For example, when $\eta = 2$ and $\beta = 1$, cooperation is not established. When both are high or both are low, cooperation is not established. With $\eta = 0.5$ and $\beta = 0.5$, around 60% cooperation is achieved, even though they are equal. This is not predicted by statements (1) and (2). It jumps to around 80% with $\eta = 0.6$ and $\beta = 0.5$, when there is a slight preference.

Comment 2

(2) Reference

In the introduction, the authors wrote about repeated PGG. Nonetheless, the authors did not cite Joshi (1987) or Boyd & Richerson (1988). If you know these papers and you chose not to cite them,

then I would like to know the reason why you did not cite them. If you did not know them simply, then please read them and think about whether you cite them or not. In addition, Takezawa& Price (2010) is also relevant.

Response

We thank you for pointing out additional citations. We have added them now.

Comment 3

(3) English

The language needs improving. I take one example, but this is just an example, and the authors have to pay more attention for the usage of English. In Introduction section, the authors wrote " In other words, after each round if each individual aware of each other' payoff, the individual adapt ~". Here, how to use "aware" is wrong. On the other hand, immediately after that, the authors wrote "However, if individuals are aware of each individual's payoff and their prestige, perhaps individuals more likely to imitate ~". Here, how to use "aware" is correct. So, it seems that the authors know how to use "aware", but the authors are just careless. I recommend that the authors should read this manuscript carefully again. If the authors make this effort, then this paper will become better.

Response

We apologize for the errors. We have done our best to eliminate or minimize them.

Appendix C

Comment on “Evolution of conditional cooperation in public good games”

General evaluation

I found many deficits after reading it. So, I cannot recommend publication as one scientist. Please see the comment below.

Major Comments

(1) I wonder whether the authors have read the papers which they cited.

In the previous referee report, I asked why the authors did not cite Joshi (1987). In the revised version, the authors cited Joshi (1987). Thank you for incorporating my comment. Thank you. When you cited Joshi (1987), you wrote, " In the literature, it has been observed that there is a possibility of graded strategies other than discrete TFT when individuals interact with many other individuals simultaneously". And there you cited Nowak & Sigmund (1992) as well as Joshi (1987). Here, I have one question. Did you read Nowak & Sigmund (1992)? Nowak & Sigmund (1992) analyzed two player games and revealed that a strategy who cooperates with a positive probability when the opponent player defected in the previous round performs well. Thus, in Nowak & Sigmund (1992)'s setting, the number of the opponent players is just one and Nowak & Sigmund (1992) examined dyadic interactions. In Nowak & Sigmund (1992)'s setting, individuals do not interact with many other individuals simultaneously. So, I wonder whether you really read Nowak & Sigmund (1992). If you still think that it is okay to cite Nowak & Sigmund (1992) in this context, then there is a possibility that you do not distinguish a population size from a group size. A population size and a group size are different. By the way, you wrote "in the literature". If you want to cite more than one paper, you should write, "in the literatures".

(2) Careless writing

I found that the authors had written this paper with little care. Please go to the reference list. Firstly, we can find Nowak (2006) twice. Reference number 9 and 36 are both Nowak (2006) published in nature. One paper should not appear twice in the reference list. In addition, in reference 36, the authors did not write in which journal Nowak (2006) was published. Secondly, we can find Panchanathan & Boyd (2004) in the reference list. The authors did not write in which journal Panchanathan & Boyd (2004) was published. I am sorry that I did not notice these mistakes in the previous version. I am very sorry. However, there are truly mistakes. So, we should correct the mistakes.

(3) English

In the previous version, I made a comment on how to usage of “aware” *as an example*. In this “specific” point, the authors improved this paper. Thank you for incorporating my comment. However, English in this paper still needs improving. Let me take one example. But it is *just an example*. We can find the expression "population imitation process is not entirely depends on the role models' payoffs and their prestige. " Here, we find two verbs. You might want to write "population imitation process *does* not entirely *depend* on the role models' payoffs and their prestige. "I recommend that the authors should read this manuscript carefully again by themselves and additionally get help by a native English speaker if possible. If the authors make this effort, then this paper will become better.

Summary

I consider that this paper does not meet the criteria of Royal Society Open Science. At least in the current form, I cannot recommend publication as one scientist. If the authors improve the quality of this paper, then it is welcome to review this paper again. I would like the authors to polish this article.

Appendix D

Response to Reviewer

Comment 1

(1) I wonder whether the authors have read the papers which they cited.

In the previous referee report, I asked why the authors did not cite Joshi (1987). In the revised version, the authors cited Joshi (1987). Thank you for incorporating my comment. Thank you. When you cited Joshi (1987), you wrote, " In the literature, it has been observed that there is a possibility of graded strategies other than discrete TFT when individuals interact with many other individuals simultaneously". And there you cited Nowak & Sigmund (1992) as well as Joshi (1987). Here, I have one question. Did you read Nowak & Sigmund (1992)? Nowak & Sigmund (1992) analyzed two player games and revealed that a strategy who cooperates with a positive probability when the opponent player defected in the previous round performs well. Thus, in Nowak & Sigmund (1992)'s setting, the number of the opponent players is just one and Nowak & Sigmund (1992) examined dyadic interactions. In Nowak & Sigmund (1992)'s setting, individuals do not interact with many other individuals simultaneously. So, I wonder whether you really read Nowak & Sigmund (1992). If you still think that it is okay to cite Nowak & Sigmund (1992) in this context, then there is a possibility that you do not distinguish a population size from a group size. A population size and a group size are different. By the way, you wrote "in the literature". If you want to cite more than one paper, you should write, "in the literatures".

Response

First, we thank the reviewer and infer that reviewer is satisfied with our previous response related to the model and interpretation of the results. We agree with reviewer's observations regarding the citation of Nowak & Sigmund (1992) citation and we regret the error. We have deleted the Nowak & Sigmund (1992) citation. We have deleted the word "literature". We have also made other grammatical changes and corrected typographical errors.

Comment 2

(2) Careless writing

I found that the authors had written this paper with little care. Please go to the reference list. Firstly, we can find Nowak (2006) twice. Reference number 9 and 36 are both Nowak (2006) published in nature. One paper should not appear twice in the reference list. In addition, in reference 36, the authors did not write in which journal Nowak (2006) was published. Secondly, we can find Panchanathan & Boyd (2004) in the reference list. The authors did not write in which journal Panchanathan & Boyd (2004) was published. I am sorry that I did not notice these mistakes in the previous version. I am very sorry. However, there are truly mistakes. So, we should correct the mistakes.

Response

We thank the reviewer and we have made appropriate changes in the references section.

Comment 3

(3) English

In the previous version, I made a comment on how to usage of “aware” as an example. In this “specific” point, the authors improved this paper. Thank you for incorporating my comment. However, English in this paper still needs improving. Let me take one example. But it is just an example. We can find the expression "population imitation process is not entirely depends on the role models’ payoffs and their prestige. " Here, we find two verbs. You might want to write "population imitation process does not entirely depend on the role models’ payoffs and their prestige. "I recommend that the authors should read this manuscript carefully again by themselves and additionally get help by a native English speaker if possible. If the authors make this effort, then this paper will become better.

Response

Thanks the reviewer. We have showed the paper to two others for checking English grammar. We have made appropriate language corrections based on their comments and feedback. Once again, we are glad that the reviewer is satisfied with the scientific content of the paper. We than the Editor and reviewer for helping us improve the paper.